# A New Synthetic Curcuminoid Displays Antitumor Activities in Metastasized Melanoma

**DOI:** 10.3390/cells12222619

**Published:** 2023-11-13

**Authors:** Leonard Kaps, Adrian Klefenz, Henry Traenckner, Paul Schneider, Ion Andronache, Rainer Schobert, Bernhard Biersack, Detlef Schuppan

**Affiliations:** 1Institute of Translational Immunology, University Medical Center, Johannes Gutenberg University Mainz, 55131 Mainz, Germany; aklefenz@students.uni-mainz.de (A.K.); htraenckner94@gmail.com (H.T.); paschnei@students.uni-mainz.de (P.S.); 2Research Center for Integrated Analysis and Territorial Management, University of Bucharest, 030018 Bucharest, Romania; ion.andronache@geo.unibuc.ro; 3Organic Chemistry 1, University Bayreuth, 95447 Bayreuth, Germany; rainer.schobert@uni-bayreuth.de; 4Division of Gastroenterology, Beth Israel Deaconess Medical Center, Harvard Medical School, Boston, MA 02115, USA

**Keywords:** black skin cancer, solid cancer, synthetic derivative of natural product, anticancer drug, curcuma longa, Ayurveda, turmeric, pulmonary metastases

## Abstract

Aim: The semisynthetic derivatives MePip-SF5 and isogarcinol, which are aligned with the natural products curcumin and garcinol, were tested for their antitumor effects in a preclinical model of pulmonary melanoma metastasis. Methods and results: MePip-SF5 was almost five times more effective in inhibiting B16F10 melanoma cell proliferation than its original substance of curcumin (IC_50_ MePip-SF5 2.8 vs. 13.8 µM). Similarly, the melanoma cytotoxicity of isogarcinol was increased by 40% compared to garcinol (IC_50_ 3.1 vs. 2.1 µM). The in vivo toxicity of both drugs was assessed in healthy C57BL/6 mice challenged with escalating doses. Isogarcinol induced toxicity above a dose of 15 mg/kg, while MePip-SF5 showed no in vivo toxicity up to 60 mg/kg. Both drugs were tested in murine pulmonary metastatic melanoma. C57BL/6 mice (*n* = 10) received 500,000 B16F10 melanoma cells intravenously. After intraperitoneal injection of MePip-SF5 (60 mg/kg) or isorgarcinol (15 mg/kg) at days 8, 11 and 14 and sacrifice at day 16, the MePip-SF5-treated mice showed a significantly (*p* < 0.05) lower pulmonary macroscopic and microscopic tumor load than the vehicle-treated controls, whereas isogarcinol was ineffective. The pulmonary RNA levels of the mitosis marker Bub1 and the inflammatory markers *TNFα* and *Ccl3* were significantly (*p* < 0.05) reduced in the MePip-SF5-treated mice. Both drugs were well tolerated, as shown by an organ inspection and normal liver- and kidney-related serum parameters. Conclusions: The novel curcuminoid MePip-SF5 showed a convincing antimetastatic effect and a lack of systemic toxicity in a relevant preclinical model of metastasized melanoma.

## 1. Introduction

Despite intense research activities, the majority of patients with advanced solid cancer have a dismal prognosis that significantly worsens when they exhibit distant (stage IV) metastases. Slowing the growth of metastases is one major therapeutic goal for achieving a stable disease [1]. This is of high importance, especially for patients with melanoma, as many of them are in a metastasized stage at their initial diagnosis [2]. Although immunotherapy and drugs that target signaling pathways have provided additional therapeutic benefits over chemotherapy alone in the last decade, the prognosis of patients in stage IV melanoma remains poor, with a 5-year survival rate of ~20% [2]. Thus, effective drugs to further improve current treatment regimens are urgently needed in the clinical setting.

Herbal medicines are rich in bioactive chemical compounds that evolved to promote pest or parasite resistance and that are actively being investigated in the search for effective and patentable chemical analogs to treat cancer or other chronic diseases [3,4].

Curcumin is a natural compound of the *Curcuma longa* rhizome which protects the rhizome from stress factors via its pronounced antioxidant activities; it can serve as a repellent of herbivores due to its bitter taste and is used as a spice in southern regions of the globe (Figure 1) [5]. In addition to its antioxidant activities, curcumin suppresses NF-κB and STAT3 signaling as well as various receptor and kinase signaling pathways [5]. A clinical Phase II trial with pancreatic cancer patients revealed responses to high-dose curcumin (8 g curcumin by mouth daily) when patients received the drug until disease progression. In two (of 21) patients, stable disease or partial response occurred and was accompanied by NF-κB suppression and a significant increase in the serum cytokine levels of the L-6, IL-8, IL-10 and IL-1 receptor antagonists [6]. Another Phase I/II study with gemcitabine-resistant pancreatic cancer patients revealed promising effects, such as a prolonged median survival (161 days; 95% confidence interval 109–223 days) and a 1-year survival rate of 19% (95% confidence interval 4.4–41.4%) [7]. Meanwhile, there are several published clinical studies that used curcumin for cancer and other diseases (such as metabolic, gastrointestinal, musculoskeletal and neuropsychiatric disorders) based on curcumin’s pleiotropic properties [8]. In Ayurveda, turmeric (*Curcuma longa*) is applied as a skin care drug, which has led to the use of curcumin especially for the chemoprevention and treatment of skin cancers [5].

However, the promising anticancer activities of curcumin are hampered by its quick metabolism to glucuronides and sulfates, low bioavailability and water solubility [9,10,11,12,13]. Due to its simple structure, the development of new (semi)synthetic curcuminoids with improved stability, uptake and bioactivities has become a promising strategy to overcome the drawbacks of using natural curcumin for cancer therapy. Synthetic fluorinated curcuminoids showed improved antitumor activity in vitro, stability, bioavailability and intestinal resorption [14]. Fluorine substitutions can play an important role in the favorable modification of drugs, as they alter the activity, conformation, pKa, membrane permeability and pharmacokinetics of drug candidates. The pentafluorothio group (SF_5_) is an electron-withdrawing, lipophilic substituent that renders negatively charged biomolecules [15]. A prominent example is the 8-pentafluorothio analog of the antimalarial mefloquine, which exhibits a several-fold-higher antimalarial in vivo activity and a longer half-life than the parent drug [15,16]. We recently described the promising effects of SF5 substituents on the antitumor activity of curcuminoids [17,18]. One derivative, named MePip-SF5 ((*E*)-1-methyl-3,5-bis(4-pentafluorothiobenzylidene)-4-piperidone), showed high antiproliferative activity against various solid tumor cell lines (including colon cancer and glioblastoma cells) that exceeded the activities of previous curcuminoids (Figure 1) [17].

Garcinol (isolated from *Garcinia indica*) and isogarcinol (obtained from garcinol under acidic conditions) are also promising natural compounds. Due to structural analogies with curcumin (e.g., a β-diketone system and phenol moiety), garcinol has shown several curcumin-type modes of action but partly exhibited higher anticancer activities than curcumin [19]. Both drugs induced apoptosis via induction of Bax, increased mitochondrial damage and TRAIL signaling and inhibited p300 histone acetyltransferase and STAT3 signaling in various cancer cells [20]. However, neither MePip-SF5 nor (iso)garcinol have been evaluated in an in vivo preclinical model of cancer or cancer metastasis. We therefore tested these derivatives in a clinically relevant mouse model of metastatic melanoma.

## 2. Materials and Methods

### 2.1. Cell Culture

3T3 murine fibroblasts (ATCC # CRL-1658), B16F10 murine melanoma cells (ATCC # CRL-6475) and RAW 264.7 murine macrophages (ATCC # TIB-71) were cultured in Dulbecco’s modified eagle’s high-glucose medium (Sigma Aldrich, Darmstadt, Germany) supplemented with 10% fetal bovine serum (FBS), 1% streptomycin–penicillin (Gibco, Darmstadt, Germany) and 1% L-glutamine (Gibco, Germany). AML12 murine hepatocytes (ATCC # CRL-2254) were cultivated in Dulbecco’s modified eagle’s F-12 Ham media supplemented as described above plus 10 µg/mL insulin, 5.5 µg/mL transferrin, 5 ng/mL selenium (all from Thermo Fisher Scientific, Braunschweig, Germany) and 40 ng/mL dexamethasone (Sigma Aldrich). All cells were maintained in a humidified incubator at 37 °C under 5% CO_2_. Cells were passaged when they reached semi-confluency. For detachment, cells were thoroughly rinsed with Dulbecco’s phosphate-buffered saline (PBS) without calcium and magnesium (Sigma Aldrich) and enzymatically treated using TrypLE™ Express (1×) (Gibco, Germany). FBS-containing medium was added to stop enzymatic activity.

### 2.2. Cytotoxicity Assay

A total of 1000 cells were seeded per well in 100 µL medium (for details, see above) using 96-well plates (Greiner Bio-On, Frickenhausen, Germany). After 24 h, curcumin, garcinol (both from Sigma Aldrich), MePip-SF5 and isogarcinol (synthesized as described previously [14,16]) were added to the wells at concentrations ranging from 0.5 to 64 µM (6 wells per concentration step) and incubated at room temperature for 48 h. For the preparation of the compounds, drugs were dissolved in DMSO at a concentration of 40 mM for stock solutions and freshly diluted with PBS to a working stock of 4 mM on the day of the experiment.

After the incubation, resazurin (Sigma Aldrich) was dissolved in fully supplemented media and added to a final concentration of 25 µg/mL per well. After incubation for 3 h, the absorbance of resorufin (7-hydroxy-*3H*-phenoxazin-3-one), the enzymatic product of resazurin, in the wells was measured on an Infinite M200Pro spectrofluorometer (TECAN, Männedorf, Switzerland) at an excitation of 570 nm. Based on the determined dose–response curves, IC_50_ concentrations were calculated by Prism 10 (GraphPad Software Version 10.1.0, La Jolla, CA, USA) using nonlinear regression analysis after conversion of concentrations to logarithmic scale.

### 2.3. Bone-Marrow-Derived Macrophages (BMDM)

Bone marrow was extracted from 8-weeks-old C57BL/6 mice (Charles River, Erkrat, Germany) and matured into primary macrophages in the presence of macrophage colony-stimulating factor (M-CSF) [21]. Briefly, mice were sacrificed by cervical dislocation, and the femurs and tibias were isolated under sterile conditions and rinsed with cold PBS supplemented with 2% heat-inactivated FBS and 1% penicillin/streptomycin using a 21G needle and a 10 mL syringe (Braun, Kronberg, Germany). Cell suspensions were passed through a 70 µm cell strainer (Milteny, Bergisch Gladbach, Germany). The remaining red blood cells were lysed with red cell lysis buffer (eBioscience, San Diego, CA, USA) and the resulting bone marrow cells resuspended in fully supplemented Iscove’s modified Dulbecco’s medium (Gibco) plus 25 ng/mL M-CSF (from a 1 mg/mL M-CSF stock solution in PBS, M-CSF, ImmunoTools, Friesoythe, Germany). Five million cells were cultured on bacterial culture plates (Sigma-Aldrich, Germany) in a total volume of 10 mL. After 4 days, half of the medium was substituted by fresh medium to yield primary macrophages after an additional 4 days.

### 2.4. Phenotype Modulation of BMDM with MePiP-SF5 and Isogarcinol

For M2-type polarization, BMDM medium was exchanged for fresh medium containing 20 ng/mL IL-4 and 20 ng/mL IL-13 (ImmunoTools, Germany), and cells were incubated for 24 h. M2-polarized macrophages were then incubated with 1.2 µM MePip-SF5, 2.75 µM curcumin, 0.65 µM isogarcinol or 1.9 µM garcinol for 24 h, corresponding to half of the IC_50_ concentration determined for RAW macrophages, followed by RNA extraction. IC_50_ concentrations were calculated as described above for the cytotoxicity assays.

### 2.5. Quantitative Reverse Transcription PCR (RT-qPCR)

Total RNA was extracted via phenol extraction from cells using RNA-Solv^®^ reagent (VWR International GmbH, Hessen, Germany) as described in the manual of the manufacturer. RNA was purified with the Monarch^®^ Total RNA Miniprep Kit (New England BioLabs GmbH, Frankfurt am Main, Germany) following the manual of the manufacturer. For RNA extraction from lung tissue, the accessory lobe of the right lung was mechanically homogenized in 1 mL RNA-Solv^®^ reagent using the Tissue Lyser II (Qiagen, Venlo, The Netherlands). A 100 µL amount of the suspension was mixed with 900 µL RNA-Solv^®^ reagent, and total RNA extraction was performed using the Monarch^®^ Total RNA Miniprep Kit as described in the manual of the manufacturer.

RNA content of the extracted solution was measured optically using the Infinite^®^ 200 PRO spectrofluorometer (TECAN, Switzerland), and 25 ng RNA per reaction was used in the following analysis. The applied primers, synthesized by Eurofins (Hamburg, Germany), are shown in Table 1.

Transcript levels of *β-Actin* were used to normalize data and to control for RNA integrity. Samples were amplified with the Luna^®^ Universal One-Step RT-qPCR Kit (New England BioLabs GmbH, Frankfurt am Main) and analyzed using a Step One Plus sequence amplification system from LifeTechnologies (Darmstadt, Germany). For normalization, results were expressed as the ratio of the copy numbers of the target gene divided by the number of copies of the housekeeping gene (*β-Actin*) within individual PCR runs.

### 2.6. Generation of Lung Metastases

Eight-weeks-old female C57BL/6 mice (body weight ~20 g) were purchased from Charles River (Sulzfeld, Germany) and kept under 12 h light–dark cycles at 25 °C and 40–60% humidity with humane care. Mice had access to regular chow and water ad libitum. Pulmonary metastases were generated by tail vein injection. A total of 500,000 B16F10 cells were resuspended in sterile 100 µL PBS (cell viability > 90%) and injected into the tail vein of the mice while animals were immobilized in a mouse restrainer (G&P Kunststofftechnik, Kassel, Germany).

### 2.7. In Vivo Toxicity and Dose Finding

Groups of 2 mice (C57BL/6) were challenged with increasing doses of MePip-SF5 or isogarcinol. Stock solutions of drugs (40 mM in DMSO, corresponding to 24 mg/mL for isogarcinol and 21 mg/mL for MePip-SF5) were freshly dissolved in PBS, yielding a total injection volume of 100 µL. Prepared drug solutions were intraperitoneally injected twice weekly (Monday and Friday) over 4 weeks. Drug doses were doubled each week, starting with 15 mg/kg body weight. Physical appearance and body weight were closely monitored. Toxic dose was defined as body weight loss of ≥10% in one week, abnormal behavior or when mice lost ≥5% of their body weight in two consecutive weeks.

### 2.8. In Vivo Antitumor Therapy

At 8, 11 and 14 days after melanoma cell injection, groups of mice (*n* = 10) received an intraperitoneal injection of isogarcinol (15 mg/kg) or MePip-SF5 (60 mg/kg) of 200 µL, while non-treated control mice received equal volumes of PBS. At day 16, mice were sacrificed, and organs (lungs, liver, kidneys and spleen) and blood were sampled for further analysis.

### 2.9. Quantification of Tumor Burden in the Lungs

Tumor nodules of the left and right lobes of the lungs were macroscopically counted. Pictures of the lungs were taken with a Stemi 2000-C binocular (Carl Zeiss, Jena, Germany), and a mm scale was pictured to create a mm grid in ImageJ software (latest version 1.54f (National Institute of Health, Bethesda, MD, USA). In addition to surface lesion counts, tumors were categorized by their size into categories of <1 mm and <2 mm. Afterwards, both lobes were formalin fixed, paraffin embedded and sectioned (5 µm) to assess intrapulmonary tumor burden. Sections were stained with ready-to-use hematoxylin and eosin staining solution (Carl Roth, Karlsruhe, Germany) for 5 min at room temperature and washed in distilled water and 0.5% acetic acid.

Per mouse, lungs were cut into 4–5 parts and then microtome thin-sectioned (6 mm). Then, 4 lung sections were randomly selected and analyzed. Sections were tile-scanned with a Leica microscope (DMi8). Pictures were binarized and the percentage of healthy vs. tumor lung tissue measured by ImageJ software using an adjusted threshold setting [22,23]. Data are expressed as means ± SD per mouse (*n* = 10 treated group; and *n* = 11 per control group).

### 2.10. Statistics

All statistical analyses were performed by using Prism 6 (GraphPad Software version 10.1.0, La Jolla, CA, USA). Data of the graphs are shown as median with 95% confidence interval. *p*-values were calculated by the Student’s *t*-test or two-way ANOVA for comparison of IC_50_ concentrations.

## 3. Results

### 3.1. In Vitro Toxicity

MePip-SF5 and isogarcinol were tested against their parent drugs and buffer controls for in vitro activity in relevant non-malignant cell lines (Table 2). MePip-SF5 showed a robust cytotoxic activity in the tested cell lines (IC_50_ 2.8 µM B16F10, 0.7 µM 3T3, 2.3 µM AML12, 1.3 µM RAW). MePip-SF5 was almost 5× more active (IC_50_ 2.8 vs. 13.8 µM, *p* < 0.0001) than its parent drug curcumin in B16F10 melanoma cells and even 10× times more active in 3T3 fibroblasts (IC_50_ 0.7 vs. 6.8 µM, *p* < 0.0001). It also showed a strong effect in RAW macrophages and AML12 hepatocytes with IC_50_ values of 1.3 and 2.3 µM, respectively. Similar to the curcuminoid MePip-SF5, isogarcinol had a stronger activity (~40%) than its original substance garcinol against B16F10 melanoma cells. Isogarcinol showed also a significantly higher cytotoxicity than garcinol in the non-malignant cell lines.

Based on these in vitro results, MePip-SF5 and isogarcinol were selected for further biological evaluation.

### 3.2. In Vivo Dose Finding

Prior to therapeutic in vivo experiments, the optimal in vivo concentration of MePip-SF5 was assessed. Mice (*n* = 2) were challenged with escalating doses of MePip-SF5 or isogarcinol, respectively. Mice tolerated a maximum single dose of 60 mg/kg MePip-SF5 or 15 mg/kg isogarcinol, respectively, without showing signs of acute toxicity such as weight loss and abnormal behavior (Figure 2).

### 3.3. Antimetastatic Effect of MePip-SF5 and Isogarcinol

MePip-SF5 and isogarcinol were tested in a murine model of pulmonary metastasis of B16F10 melanoma cells. After establishment of metastases, mice received intraperitoneally three injections of MePip-SF5 (60 mg/kg) or isogarcinol (15 mg/kg) on days 8, 11 and 14 after tumor cell inoculation (Figure 3A). At sacrifice on day 16, mice treated with MePip-SF5 showed significantly (*p* < 0.05) lower numbers of macroscopic tumor nodules of size ≤ 1 mm in the lungs compared to controls and isogarcinol-treated mice (Figure 3B). Central safety laboratory parameters for cytolysis (LDH) and kidney (creatinine, urea) and liver (ALT, AST, GLDH, γ-GT, Bilirubin, ALP) injury remained normal in all studied groups, without statistical differences, indicating that MePip-SF5 and isogarcinol at the applied doses were well tolerated (Appendix A). Organ weights remained normal, and macroscopic inspections of major organs (kidneys, liver and spleen) displayed neither metastatic nodules nor signs of overt toxicity. Further, body weights did not change under either treatment (Appendix A).

### 3.4. Anti-Inflammatory and Antimitotic Effect of MePip-SF5

Expression of relevant inflammatory and cell-cycle-related transcripts were quantified in lung tissues to assess potential mechanisms of action of MePip-SF5 in vivo. Expression of *Bub1*, a mitosis marker, was significantly downregulated in lungs of MePip-SF5-treated mice. This was accompanied by a trend of reduced expression of the cell proliferation marker *Ki67* in the MePip-SF5-treated group. The apoptosis markers *Bcl2* and *Bax* remained unchanged, while MePip-SF5 treatment reduced the inflammatory markers *Tnfa* and *Ccl3* in metastatic lungs and did not affect the inflammatory transcripts *Il6* and *Hif1a* (Figure 4).

### 3.5. Immunomodulatory Effect of MePip-SF5 on Primary Macrophages

MePip-SF5 showed a significant antitumor effect in mice with pulmonary melanoma metastasis. The curcuminoid reduced transcript levels of the inflammatory cytokines *TNFα* and *Ccl3*, which are involved in general immune cell activation and upregulation of the mitosis marker *Bub1* in metastatic lungs. We sought to clarify whether the immune-modulating effect of MePip-SF5 can also be observed in macrophages, which are important immune regulatory cells that can both support and suppress anticancer T cell responses [24,25,26].

Therefore, MePip-SF5 and curcumin were tested in M2-polarized primary macrophages that suppress anticancer immune responses (Figure 5). Drug doses were chosen based on our toxicity data for RAW macrophages (Figure 1). With an effective dose of 0.6 µM, MePip-SF5 increased the expression of the proinflammatory cytokine *Ccl2*, which orchestrates the recruitment and activation of inflammatory cells such as polymorphonuclear leukocytes [27]. In contrast, MePip-SF5 had no effect on the expression of *Inos*, which is mainly produced by macrophages in defense against bacterial infection (Figure 5). *Ccl3* expression was also increased by isogarcinol but not by garcinol (Figure 5). MePip-SF5 upregulated *Ccl3* in vitro; however, its pulmonary expression levels were downregulated in MePip-SF5-treated animals with melanoma metastasis. The other mRNA expression data also suggest that immunomodulatory effects are not the primary mechanisms of action of MePip-SF5.

## 4. Discussion

We have tested the in vivo antitumor activity of the curcumin derivative MePip-SF5 and isogarcinol in a relevant preclinical model of pulmonary metastatic melanoma. Both drugs showed promising cytotoxic in vitro effects on different cancer cell lines, including melanoma, that warrant further biological evaluation in vivo [28,29,30,31].

Melanoma is the most aggressive form of skin cancer and can easily metastasize from a relatively small primary tumor to multiple sites, including lung, liver, brain, bone and lymph nodes. Thus, therapeutic strategies to prevent or stop the growth of metastases to support stable disease, shrinkage of lesions or even a partial or complete response are urgently needed. Natural products are a vital source of new drugs to fight cancer. Up to 60% of the anticancer drugs approved during the last 30 years are related to natural products [32]. To date, relatively few structurally defined natural products have been translated into approved anticancer drugs. However, these unique molecules can serve as a structural template for the design of more potent anticancer agents.

MePip-SF5 is a good example for this strategy as its chemical structure is based on curcumin, which is a bright-yellow natural product also used for food flavoring/coloring in the Asian kitchen and a molecule with established antioxidant, anticancer and immune-modulatory properties (Figure 1) [5]. In addition, curcumin promotes the M2 phenotype of macrophages and, thus, modulates inflammation [33]. The chemical modification of curcumin to MePip-SF5 resulted in an improved cytotoxicity towards cancer cells [14]. In our present study, IC_50_ values were almost 5× lower for MePip-SF5 than for its parent compound curcumin in B16F10 melanoma cells, while the other tested compound isogarcinol was almost 40% more active than garcinol (Table 1). When both derivatives were tested in the in vivo model of metastatic lung cancer using B16F10 melanoma cells, MePip-SF5 showed a robust antitumor effect, while isogarcinol failed to show activity (Figure 3). This underlines the fact that in vitro experiments cannot entirely predict efficacy of drugs in vivo, considering both drugs exhibited a comparable activity in vitro in the melanoma cells (MePip-SF5 2.8 µM vs. isogarcinol 2.1 µM) [34]. One major weakness of in vitro experiments is that they cannot replicate the behavior of the drug in the complex mammalian organism where, e.g., pharmacokinetics/-dynamics, biocompatibility and intercellular interactions, especially of the immune system and the cancer cells, are difficult if not impossible to replicate in vitro. The drug needs to circulate long enough in the bloodstream to reach an effective dose at its target side, avoiding rapid clearance by kidneys and liver (enzymatic degradation) and the effect of too-high binding to blood proteins [35]. Second, it needs to penetrate the tumor tissue including the tumor microenvironment. For small-molecule drugs, polarity and active transport are keys for efficient uptake and diffusion of the drug out of the vessels into the tumor. Third, a wide therapeutic window of a drug is an important factor to overcome their preclinical stage.

MePip-SF5 was less toxic than isogarcinol and could be dosed 8× more highly than isogarcinol in our model, which certainly rendered a therapeutic advantage considering both drugs seemed to be equal potent in vitro (Figure 2). However, laboratory safety and biometric parameters of body and organ weight suggested that both drugs were well tolerated by the mice, and none of the drugs was overdosed (Appendix A). Notably, MePip-SF5 was clearly an active anticancer/antimetastatic agent in vivo compared to the ineffective isogarcinol when used at doses that exhibited comparable anticancer cell effects in vitro, which highlights the relevance of in vivo confirmation of in vitro drug effects. Therefore, MePip-SF5 is a novel curcuminoid with anticancer properties like piperidin-4-one-based curcuminoids such as EF24 and DiFiD [36,37,38].

Analysis of inflammatory markers in the cancerous lungs indicated that the mechanism of action of MePip-SF5 might be based on an (indirect) immune-modulatory effect. We were not able to demonstrate a direct effect of the drug on key immune cells in vitro since, e.g., the inflammatory cytokine Ccl3 (mainly produced by macrophages) was downregulated in MePip-SF5-treated mice but upregulated in M2-polarized (tumor-associated) macrophages in vitro (Figure 4 and Figure 5). The major anticancer/antimetastatic effect of MePip-SF5 may therefore be related rather to interference with the cell cycle of the rapidly proliferating melanoma cells because Bub-1, an important negative regulator of the cell cycle, was downregulated by MePip-SF5 treatment (Figure 4).

Our study has limitations that have to be acknowledged and can be addressed in follow-up studies. First, we tested the compounds only in one mouse model for pulmonary metastasis of melanoma. Whether MePip-SF5 has an effect on B16F10 metastases in other organs (e.g., liver) needs to be proven. Second, the therapeutic relevance of MePip-SF5 for other solid tumors (e.g., pancreatic or liver cancer) has to be evaluated. Third, additional studies need to be undertaken to reveal the mechanism of MePip-SF5 (e.g., by transcriptomics). Fourth, based on our promising in vivo data, MePip-SF5 could be a valuable addition to established regimes, e.g., in combination with immunotherapy, and merits studies using such combinations. Fifth, we did not measure protein levels of the corresponding inflammatory transcripts and apoptosis markers, which might weaken our conclusion as mRNA levels do not entirely reflect corresponding protein levels [39]. This accounts especially for the apoptosis markers Bcl2 and Bax, which undergo posttranslational modification.

In conclusion, we have tested the novel curcumin derivative MePip-SF5 for its anticancer and antimetastatic efficacy in a relevant clinical model of metastasized melanoma for the first time. The drug showed a convincing therapeutic effect and did not show signs of acute toxicity, which deserves further investigation.

## Figures and Tables

**Figure 1 cells-12-02619-f001:**
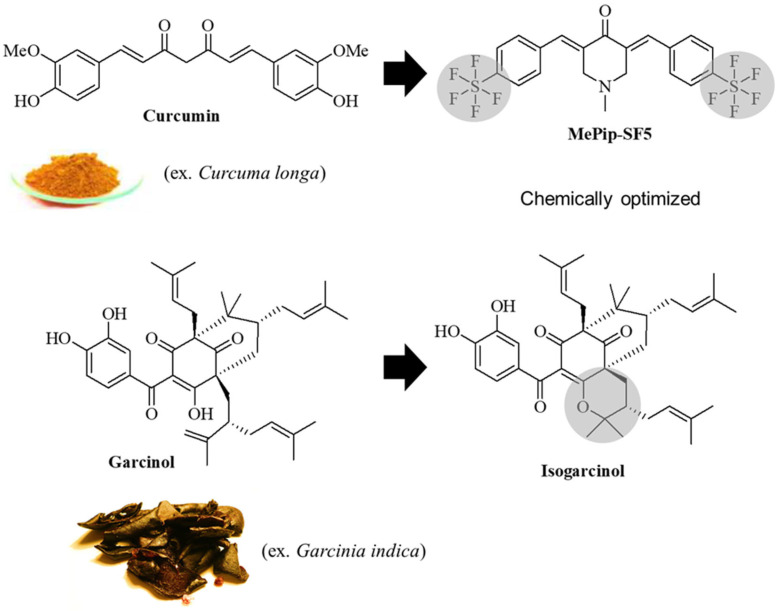
Structures of curcumin, its synthetic derivative MePip-SF5 and (iso-)garcinol.

**Figure 2 cells-12-02619-f002:**
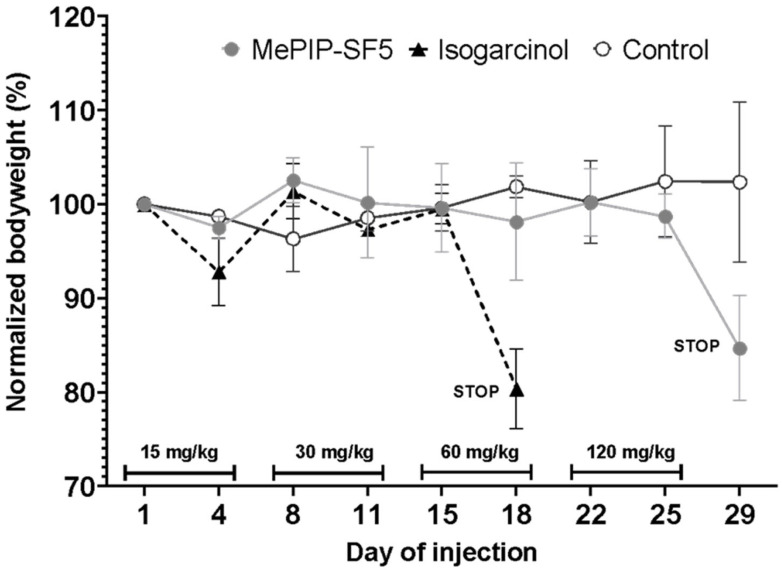
In vivo toxicity of MePIP-SF5 and isogarcinol. The body weight of the mice was closely monitored as they were challenged twice weekly with escalating doses of MePIP-SF5 or isogarcinol. The experiment was discontinued when mice experienced acute toxicity as indicated by ≥10% weight loss in one week and/or abnormal behavior.

**Figure 3 cells-12-02619-f003:**
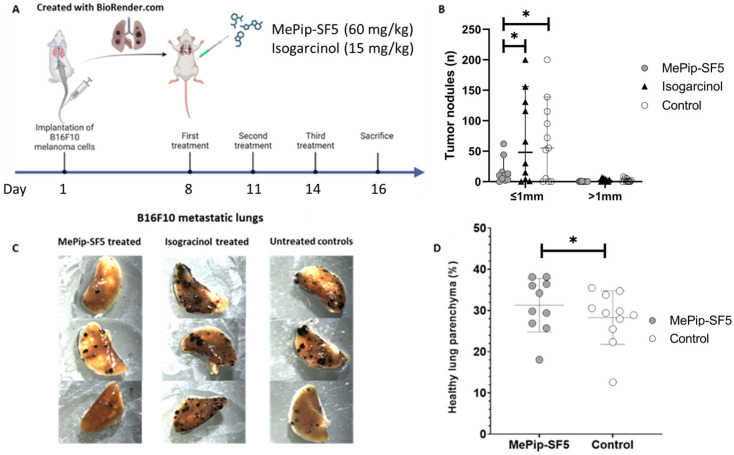
In vivo treatment of B16F10 metastatic melanoma with MePip-SF5 and isogarcinol. (**A**) Treatment scheme: mice were injected intraperitoneally with MePip-SF5 (60 mg/kg) or isogarincol (15 mg/kg) on days 8, 11 and 14 after tumor cell inoculation. (**B**) Quantification of macroscopic tumor load on the surface of lungs. (**C**) Representative pictures of lungs from mice treated with MePip-SF5 or isogarcinol and PBS-treated control mice. (**D**) Morphometric quantification of healthy lung parenchyma of thin sections of lungs demonstrating a significantly higher proportion of healthy lung tissue in mice treated with MePip-SF5 than in PBS-treated controls (*n* = 10 for treated groups and *n* = 11 for controls, * *p* < 0.05).

**Figure 4 cells-12-02619-f004:**
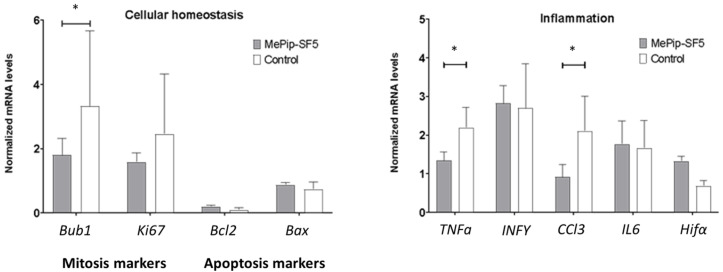
Quantification of transcript levels of cellular homeostasis and inflammation in lung tissue of MePIP-SF5-treated and control mice (* *p* < 0.05).

**Figure 5 cells-12-02619-f005:**
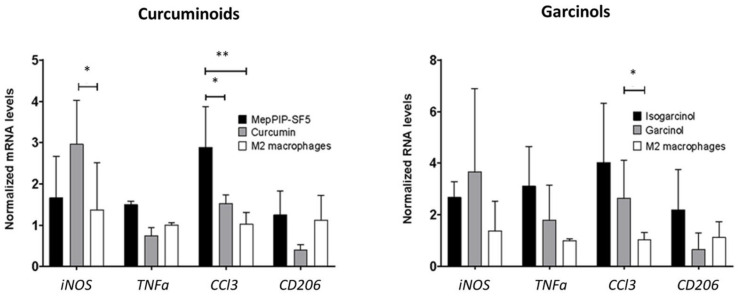
Immunomodulatory effect of MePip-SF5, curcumin, isogarcinol and garcinol on M2-polarized bone-marrow-derived macrophages. Cells were incubated with MePip-SF5 (1.2 µM), curcumin (2.75 µM), isogarcinol (0.65 µM) or garcinol (1.9 µM) for 48 h and transcript level determined by qRT-PCR normalized to *β-Actin* transcripts (values are means ± SD from 3 replicates; * *p* < 0.05; ** *p* < 0.01).

**Table 1 cells-12-02619-t001:** Primer sequences used for RT-qPCR.

	Forward Primer	Reverse Primer
*Tnf-a*	CCTGTAGCCCACGTCGTAG	GGGAGTAGACAAGGTACAACCC
*iNos*	GTTCTCAGCCCAACAATACAAGA	GTGGACGGGTCGATGTCAC
*Ccl3*	TTCTCTGTACCATGACACTCTGC	CGTGGAATCTTCCGGCTGTAG
*Mrc1*	AAGGCTATCCTGGTGGAAGAA	AGGGAAGGGTCAGTCTGTGTT
*Bcl2*	TACCGTCGTGACTTCGCAGAG	GGCAGGCTGAGCAGGGTCTT
*Bax*	AGACAGGGGCCTTTTTGCTAC	AATTCGCCGGAGACACTCG
*Ki67*	ATCATTGACCGCTCCTTTAGGT	GCTCGCCTTGATGGTTCCT
*Bub1*	ATGCAAAGCTACACGGGTAATG	GGTCACTGTTGTACTCAGCAAA
*Hif1-*α	CGACACCATCATCTCTCTGG	TGATTCAGTGCAGGATCAGC
*IL-6*	CTGCAAGAGACTTCCATCCAG	AGTGGTATAGACAGGTCTGTTGG
*INF-γ*	CGGCACAGTCATTGAAAGCC	TGCATCCTTTTTCGCCTTGC
*β-Actin*	GGCATTGTTACCAACTGGGACGAC	CCAGAGGCATACAGGGACAGCACAG

**Table 2 cells-12-02619-t002:** IC_50_ values of MePip-SF5, isogarcinol and their corresponding parent compounds.

Cell Lines	Curcumin	MePip-SF5	*p*-Value (Curcumin vs. MePip-SF5)	Garcinol	Isogarcinol	*p*-Value (Garcinol vs. Isogarcinol)	*p*-Value (MePip-SF5 vs. Isogarcinol)
	µM (95% Confidence Interval)						
B16F10 melanoma cells	13.8	2.8	*p* < 0.0001	3.1	2.1	*p* < 0.0001	*p* < 0.0001
	(12.31–15.5)	(2.087–3.661)		(2.664–3.694)	(1.824–2.369)		
3T3 fibroblasts	6.8	0.7	*p* < 0.0001	8.7	3.2	*p* < 0.0001	*p* < 0.0001
	(6.246–7.423)	(0.711–0.804)		(7.112–10.79)	(2.959–3.555)		
AML12 hepatocytes	10.8	2.3	*p* < 0.0001	10.1	4.3	*p* < 0.0001	*p* < 0.0001
	(9.421–12.46)	(1.506–3.51)		(9.023–11.51)	(3.326–5.572)		
RAW macrophages	5.5	1.3	*p* < 0.0001	3.8	2.3	*p* < 0.0001	*p* < 0.0001
	(5.017–6.065)	(1.024–1.517)		(3.275–4.368)	(2.058–2.489)		

## Data Availability

Raw data are available on reasonable request.

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
