# Peer review of "A New Synthetic Curcuminoid Displays Antitumor Activities in Metastasized Melanoma"

_cells, 2023, doi:10.3390/cells12222619_

Round 1
Reviewer 1 Report
Comments and Suggestions for Authors
The scientific idea behind the manuscript is very interesting and the investigations carried along with data presentation is good. However, I believe that the manuscript requires some revision.
In your study, use of garcinol and isogarcinol as comparators is quite weak. The fact that they may be structurally related to curcumin can be challenged. Please provide sufficient evidence to support your argument.
There are a few grammatical/typographical errors in the manuscript and places where you need to provide more information.
Line 66: delete “the” in front of the word “pleotropic”.
Line 121: how were the drug solutions prepared? Please provide further details
Line 125: “absorption of the wells was measured”. Please clarify what you are measuring here.
Line 144 to 146: How were the solutions prepared and how was IC50 determined? Please provide further information.
Line 179: You state that the drugs were prepared in “PBS substituted with 10% FBS”. These drugs are quite lipophilic. How were they dissolved in an aqueous solution and at what concentrations?
Author Response
Manuscript ID: cells-2707830
"A New Synthetic Curcuminoid Displays Antitumor Activities in Metastasized Melanoma"
Dear Professor Vestek,
We would like to thank you for returning the reviewers’ comments and providing us the opportunity to resubmit after addressing the points raised by the reviewers. The revised manuscript contains all changes highlighted and a point-to-point response is included below. We believe that the comments and subsequent changes have improved the manuscript and hope that it can further be considered for publication.
Please find the corrected version of the manuscript attached.
Best regards,
Leonard Kaps MD PhD, Bernhard Biersack PhD and Detlef Schuppan MD PhD
Reviewer 1
We thank the reviewer for the given suggestions and thoughtful comments. The suggested corrections have significantly improved the manuscript.
Comments and Suggestions for Authors
Kaps et al. describe the effects of two new synthetic compounds derived respectively from curcuma and garcinol on metastasized melanoma. They have mainly performed their analyses by measuring the mRNA levels instead of the protein accumulation. The article is interesting as it potentially identifies a new effective drug that could be used for metastatic tumors, such as melanoma. Despite this positive comment, the authors need to respond to the following comments to clarify some points, mainly in the results part.
- In the table 2, IC50 values are indicated without any standard deviation. It is thus difficult to confirm the significance of some assertions in the comparison of the drugs. E.g., is the difference between garcinol (3.1 microM) and isogarcinol (2.1 microM) significant? Statistical tests should be performed for each IC50 to be able to conclude, even for curcumin vs. MePip-SF5.
We thank the reviewer for this thoughtful comment and agree that p-values were missing in table 2. We have added this information to table 2 and do also the comparison between MePip-SF5 vs. isogarcinol.
- Conversely to table 2, Figure 2 shows standard deviations in the graphs while N=2. How is it possible? This should be corrected.
We agree and have corrected the graph, now showing both mice.
3. Figure 3B is unclear in its building: for the tumors <2 mm (is it really <2, by the way?) the bars for the animals treated by MePip-SF5 do not seem related to anything. Please correct that point. Besides, it is not clear how tumor quantification was performed: the samples within the "total" group do not seem to be equal to "<1mm" group + "<2mm" group. Please explain that or correct the figure.
We have corrected the graph and apologize for this inattention. We have removed the “total” quantification as it is indeed difficult to interpret without having the exact numbers. We hope that is now clear.
4. The authors have measured the levels of Bub1, Ki67, Bcl2 and Bax, at the transcript level. However, Bcl2 and Bax need to be modified post-translationnaly. The make the comment short, it is the accumulation of the protein that is really informative not its mRNA. The authors should deeply justify why they have measures transcript levels and how this decreased their conclusions. The same comments could be done for the inflammatory markers (Fig4 - right- and Fig 5). Western blot (for the in vivo experiments) or immunological assay in the medium (macrophages experiments) should be preferred and performed.
We agree that mRNA levels do not necessarily reflect protein levels as demonstrated by several high-quality studies (e.g. https://www.cell.com/cell/pdf/S0092-8674(16)30270-7.pdf). However, there is solid evidence that differentially expressed mRNAs correlate significantly better with their protein product than non-differentially expressed mRNAs (https://www.nature.com/articles/srep10775). We have added this limitation in the discussion and now clearly state that we were not able to measure protein levels especially for Bcl2 and Bax.
We hope that this is acceptable for you.
Reviewer 2
We thank the reviewer for her/his careful correction of the manuscript and thoughtful comments. We have corrected the manuscript, adapting the suggestions.
The scientific idea behind the manuscript is very interesting and the investigations carried along with data presentation is good. However, I believe that the manuscript requires some revision.
In your study, use of garcinol and isogarcinol as comparators is quite weak. The fact that they may be structurally related to curcumin can be challenged. Please provide sufficient evidence to support your argument.
There are a few grammatical/typographical errors in the manuscript and places where you need to provide more information.
1. Line 66: delete “the” in front of the word “pleotropic”.
Corrected
2. Line 121: how were the drug solutions prepared? Please provide further details
We have added this information in 2.2 “cytotoxic assay”.
3. Line 125: “absorption of the wells was measured”. Please clarify what you are measuring here.
We agree that the method was insufficiently described. We have improved it accordingly.
4. Line 144 to 146: How were the solutions prepared and how was IC50 determined? Please provide further information.
We have added this information accordingly.

Reviewer 2 Report
Comments and Suggestions for Authors
Kaps et al. describe the effects of two new synthetic compounds derived respectively from curcuma and garcinol on metastasized melanoma. They have mainly performed their analyses by measuring the mRNA levels instead of the protein accumulation. The article is interesting as it potentially identifies a new effective drug that could be used for metastatic tumors, such as melanoma. Despite this positive comment, the authors need to respond to the following comments to clarify some points, mainly in the results part.
* In the table 2, IC50 values are indicated without any standard deviation. It is thus difficult to confirm the significance of some assertions in the comparison of the drugs. E.g., is the difference between garcinol (3.1 microM) and isogarcinol (2.1 microM) significant? Statistical tests should be performed for each IC50 to be able to conclude, even for curcumin vs. MePip-SF5.
* Conversely to table 2, Figure 2 shows standard deviations in the graphs while N=2. How is it possible? This should be corrected.
* Figure 3B is unclear in its building: for the tumors <2 mm (is it really <2, by the way?) the bars for the animals treated by MePip-SF5 do not seem related to anything. Please correct that point. Besides, it is not clear how tumor quantification was performed: the samples within the "total" group do not seem to be equal to "<1mm" group + "<2mm" group. Please explain that or correct the figure.
* The authors have measured the levels of Bub1, Ki67, Bcl2 and Bax, at the transcript level. However, Bcl2 and Bax need to be modified post-translationnaly. The make the comment short, it is the accumulation of the protein that is really informative not its mRNA. The authors should deeply justify why they have measures transcript levels and how this decreased their conclusions. The same comments could be done for the inflammatory markers (Fig4 - right- and Fig 5). Western blot (for the in vivo experiments) or immunological assay in the medium (macrophages experiments) should be preferred and performed.
Author Response

(The authors gave the same response as above.)

Round 2
Reviewer 2 Report
Comments and Suggestions for Authors
Authors have adequately answered the comments.